# A Novel Magnetic Fluorescent Fe$_3$O$_4$@ZnS@MPS Nanosensor for Highly Sensitive Determination and Removal of Ag$^+$

Yan Gao [1], Xin Chen [2], Ping Xu [2], Jie Chen [1], Shihua Yu [2], Zhigang Liu [1],* and Xiaodan Zeng [1],*

1   Center of Characterization and Analysis, Jilin Institute of Chemical Technology, Jilin 132013, China; gyaxz@126.com (Y.G.); jiechendr@163.com (J.C.)
2   School of Chemical and Pharmaceutical Engineering, Jilin Institute of Chemical Technology, Jilin 132013, China; chenxin92@jlict.edu.cn (X.C.); xuping@jlict.edu.cn (P.X.); ysh@jlict.edu.cn (S.Y.)
*   Correspondence: lzg@jlict.edu.cn (Z.L.); jiangzxd@jlict.edu.cn (X.Z.)

**Abstract:** A novel magnetic fluorescent nanoprobe (Fe$_3$O$_4$@ZnS@MPS(MFNPs)) was synthesized, which recognized and cooperated with Ag$^+$ ions, and a rapid method for detecting Ag$^+$ was established in solution. It was found by fluorescence spectroscopy analysis that the MFNPs could detect Ag$^+$ in PBS solution and, upon addition of Ag$^+$ ions, the fluorescence (FL) of MFNPs could be quenched significantly. The sensor has a low limit of detection (LOD) of 7.04 μM for Ag$^+$. The results showed that MFNPs were extremely specific and sensitive for the quantitative detection of Ag$^+$ over a wide pH range. Then, the recognition mechanism between MFNPs and guest Ag$^+$ was explored via measures of infrared spectroscopy and electron microscopy. It was speculated that the oxygen atoms in the sulfonic acid group cooperated with Ag$^+$ to form a synergistic complexation. The assay was successfully used to determine the content of Ag+ in real samples.

**Keywords:** magnetic fluorescent nanosensor; selective recognition; silver ion; 3-sulfhydryl-1-propane sodium (MPS)





## 1. Introduction

Silver has always attracted attention due to its unique chemical properties of strong corrosion resistance and high antioxidant capacity. Because of its rarity and high gloss, silver is widely used in the production of daily necessities [1,2]. A large number of widespread uses have a relatively significant impact on the environment. Therefore, it is necessary to adopt appropriate detection methods to analyze it. Excessive human exposure to silver is likely to cause silver poisoning, growth retardation, etc., and too much silver hurts the eyes and skin [3,4]. Thus, it is of great significance to investigate an effective method for detection of Ag$^+$.

Silver ions can be detected by a variety of methods, such as fast but unstable electrochemical detection and accurate but expensive flame atomic absorption spectrometry [5–7]. In addition to these techniques, the extraction method using molecular receptors or chelating ligands is also widely used in the detection of Ag(I). Compared with conventional assays, fluorescence detection has greater advantages, such as rapid reaction, simple operation, low cost, and high sensitivity [8–10].

However, these fluorescent probes also cause further pollution to the environment when they are used to detect Ag$^+$. Compared with these fluorescent probes, ZnS QDs have unique advantages, such as a narrow and symmetric emission spectrum, excellent light stability, and resistance to photobleaching [11]. Therefore, the development of multifunctional fluorescence chemosensors for sensing of heavy metals, that are friendly to the environment, is increasingly attracting attention [12].

In recent years, magnetic nanomaterials have attracted significant attention in the academic field due to their excellent magnetic reaction ability, high biocompatibility, and good stability [13–15]. Among these, superparamagnetic iron oxide (Fe$_3$O$_4$) is particularly

versatile due to its unique high coercivity, easily functional modification, excellent controllable magnetic responsiveness, which can be manipulated by external magnetic fields, and controllable size and surface [16–18]. This enables researchers in various fields, such as chemistry, biology, medicine, and materials science, to use MFNPs to construct multifunctional nanoprobes and conduct significant research in sewage treatment, biological imaging, drug delivery, and other fields [19,20].

However, due to the strong magnetic dipole attraction between the particles, $Fe_3O_4$ nanoparticles tend to aggregate. Therefore, stabilizers such as organic matter, semiconductors, and oxides with specific functional groups are often applied to the nanoparticles' surface in order to improve stability. Using a variety of biocompatible polymers to functionalize and modify the surface of $Fe_3O_4$ nanoparticles to provide new functions is the focus of current research. When the stability of $Fe_3O_4$ nanoparticles is guaranteed, the introduction of fluorescent substances is conducive to the separation and transfer of detection substances, especially heavy metals in water, thus expanding the application field of fluorescent probes. Aggregation-induced sedimentation technology can efficiently and rapidly remove $Ag^+$ from water samples and can avoid secondary pollution [21].

In this work, a novel environmentally friendly magnetic fluorescent nanosensor ($Fe_3O_4$@ZnS@MPS(MFNPs)) modified with 3-sulfhydryl-1-propane sodium for coinstantaneous detection and removal of $Ag^+$ from water samples is reported (Scheme 1). The results showed that MFNPs achieved highly specific recognition and were extremely sensitive for the quantitative detection of $Ag^+$ over a wide pH range. The sensor has a low limit of detection (LOD) of 7.04 µM for $Ag^+$. The optimal adsorption capacity of MFNPs was calculated to be about 395.79 mg/g and the optimal adsorption rate capacity of MFNPs was calculated to be about 98%. This work provides a new method for the synthesis of magnetic fluorescent nanosensors, exploiting their application not only for the detection of $Ag^+$, but also other heavy metal ions, with the multiple functions of enrichment, detection, and separation.

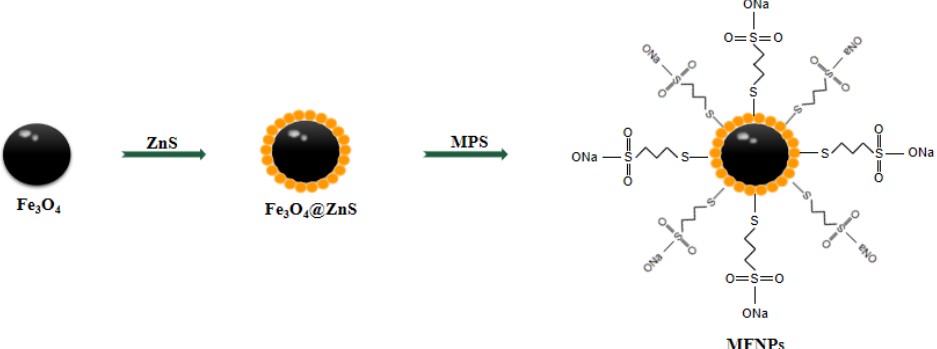

**Scheme 1.** Fabrication of MFNPs.

## 2. Materials and Methods

### 2.1. Preparation of Fe₃O₄ Magnetic Nanoparticle

The magnetic $Fe_3O_4$ nanoparticle was synthesized via the solvothermal method and prepared as follows: $FeCl_3 \cdot 6H_2O$ (2.7 g) was dissolved in 60 mL ethylene glycol in a water bath and stirred until fully dissolved. Sodium acetate (7.2 g) and surfactant (0.5 g) were added and stirred for 30 min. The solution was moved to a Polytetrafluoroethylene kettle, which was sealed tightly, and the hydrothermal reaction was carried out at high temperature. The reaction was then allowed to end and to cool naturally. The solution was moved to a Teflon kettle, which was sealed, followed by a hydrothermal reaction at a high temperature and natural cooling at the end of the reaction. The black precipitate was collected after washing several times, and the dispersive solution was prepared by adding water for further modification.



### 2.2. Preparation of Fe₃O₄@ZnS

The chemical coprecipitation method was employed to synthesize $Fe_3O_4$@ZnS core/shell nanocomposites. First, 10 mL of $Fe_3O_4$ was diluted to 100 mL and the solution was kept neutral. Then, 2.5 mmol of $Zn(Ac)_2 \cdot 2H_2O$ and $Na_2S \cdot 9H_2O$ were dissolved in turn under the condition of water bath stirring, and were stirred for 6 h. The item was attracted by a magnetic force and then underwent a thorough cleansing process with ultrapure water and ethanol. The $Fe_3O_4$@ZnS nanoprobe was completed and water was added to make a dispersion for further modification [22].

### 2.3. Preparation of Fe₃O₄@ZnS@MPS

A quantity of 5 mL of $Fe_3O_4$@ZnS ethanol solution was placed in a round-bottom flask. Then, 5 mL 3-sulfhydryl-1-propane sodium (MPS) (89.105 mg, 0.1 mmol/L) ethanol solution was added. The mixture was then stirred in a water bath at 40 °C for 4 h in the dark. After the item was attracted by a magnetic force, it underwent a thorough cleansing process with ultrapure water and ethanol. Water was added to the prepared $Fe_3O_4$@ZnS@MPS microspheres to make a dispersion solution.

### 2.4. Instrumentation

The surface topography of the microspheres was observed via a scanning electron microscope (SEM) (Quanta 200, FEI, Hillsboro, OR, USA) (EHT = 5.00 kV) and a transmission electron microscope (TEM) (JEM 2100, JEOL, Tokyo, Japan) (Accelerating Voltage = 200 kV). Fourier transform-infrared (FTIR) spectra were obtained from a spectrometer (Thermo Scientific Nicolet IS50, Thermofisher, Waltham, MA, USA). X-ray diffraction (XRD) images were obtained using an X-ray diffractometer (D8FOCUS, Bruker, Berlin, German). The measurement was conducted using CuK$\alpha$ radiation ($\lambda = 1.5406$ Å) in the range of $2\theta = 20°$–$80°$ at a scan speed of $2°$/min. XPS spectra data were obtained on an X-ray photoelectron spectrometer (Krayos AXIS Ultra DLDX, Shimadzu, Tokyo, Japan) (Analyzer Mode: Constant Analyzer Energy: Pass Energy 30.0 eV; Energy Step Size: 0.100 eV). The magnetism (VSM) was measured using a vibrating sample magnetometer (7404 vibrating sample magnetometer, LakeShore, Carson, CA, USA). The thermogravimetric (TG) curves were obtained using a thermalgravimetric analyzer (Discovery SDT650 synchronous TGA-DTA Instrument, TA, Los Angeles, CA, USA). Fluorescence spectra were measured on a fluorescence spectrophotometer (FluoroMax-plus instrument, Horiba, Austin, TX, USA) with an excitation of (370) nm. The concentration of $Ag^+$ in solution was determined by atomic absorption spectrometry (AAS) (ICP-OES:Thermo Fisher iCAP 7400, Thermofisher, USA).

### 2.5. Materials

Other chemicals used in this study, such as $FeCl_3 \cdot 6H_2O$, $CH_3COONa \cdot 3H_2O$, $Zn(ac)_2$, $Na_2S$, polyethylene glycol 2000, 3-sulfhydryl-1-propane sodium, $Co^{2+}$, $Pb^{2+}$, $Ni^{2+}$, $Hg^{2+}$, $Al^{3+}$, $Cu^{2+}$, $Zn^{2+}$, $Cd^{2+}$, $Fe^{3+}$, $Fe^{2+}$, $K^+$, $Ca^{2+}$, and NaCl, were purchased from Shanghai Aladdin Bio-Chem Technology Co., Ltd. (Shanghai, China).

### 2.6. Procedure for the Fluorescent Detection of Ag⁺

Certain volumes of $Ag^+$ solutions were added to 0.4 mg of the $Fe_3O_4$@ZnS@MPS suspension, whose final volume was tuned to 5 mL by PBS buffer solution. After a 5 min reaction, the emission spectra were measured at an excitation wavelength of 370 nm. The selectivity performance of $Fe_3O_4$@ZnS@MPS assay for the detection of $Ag^+$ was carried out by measuring its fluorescence response in the presence of various common ions ($Co^{2+}$, $Pb^{2+}$, $Ni^{2+}$, $Hg^{2+}$, $Al^{3+}$, $Cu^{2+}$, $Zn^{2+}$, $Cd^{2+}$, $Fe^{3+}$, $Fe^{2+}$, $K^+$, $Ca^{2+}$, $Na^+$).

### 2.7. Removal and Adoption Capacity Performance of Ag⁺

The removal and adoption capacity performance of $Ag^+$ testing was determined in a centrifuge tube, for different concentrations of $AgNO_3$, after shaking well.

The removal activities can be evaluated by the following equation:

$$R = \frac{C_0 - C_1}{C_0} \times 100\%$$ (1)

$C_1$ is the final concentration of $Ag^+$, and $C_0$ is the initial concentration of $Ag^+$ ions. The concentration of $Ag^+$ ions was analyzed via AAS.

The adsorption amount of $Ag^+$ (Q mg/g) can be calculated according to the following equation:

$$Q_e = \frac{(C_0 - C_1)V}{m}$$ (2)

$Q_e$ is the adsorption capacity ($mg \cdot g^{-1}$), $C_0$ and $C_e$ are the initial and equilibrium concentrations of $Ag^+$ ($mg \cdot L^{-1}$), respectively, V is the volume of the metal ion solution (L), and m is the $Fe_3O_4@ZnS@MPS$ mass (g).

## 3. Results and Discussion

### 3.1. Characterization of MFNS

#### 3.1.1. Morphology Analysis

The morphology and particle size of $Fe_3O_4$ MNPs and $Fe_3O_4@ZnS@MPS$ core-shell nanocomposites were investigated using SEM and TEM.

Figure 1a,c show the SEM and TEM images of $Fe_3O_4$ MNPs, from the commercial $Fe_3O_4$ with a diameter distribution from 150 to 250 nm, which reveal that $Fe_3O_4$ MNPs have a regular and uniform distribution. In addition, the SEM image of $Fe_3O_4@ZnS@MPS$ is shown in Figure 1b. Compared with Figure 1a,b, it can be seen that ZnS particles are accumulated on the surface of $Fe_3O_4$ after the modification of the thiol group, and the surface of $Fe_3O_4$ MNPs becomes irregular and rough. In addition, TEM images (Figure 1d) of $Fe_3O_4@ZnS@MPS$ with a diameter distribution from 200 to 300 nm were obtained, which show that ZnS particles modified with sulfhydryl groups have been deposited on the surface of $Fe_3O_4$ MNPs [23].

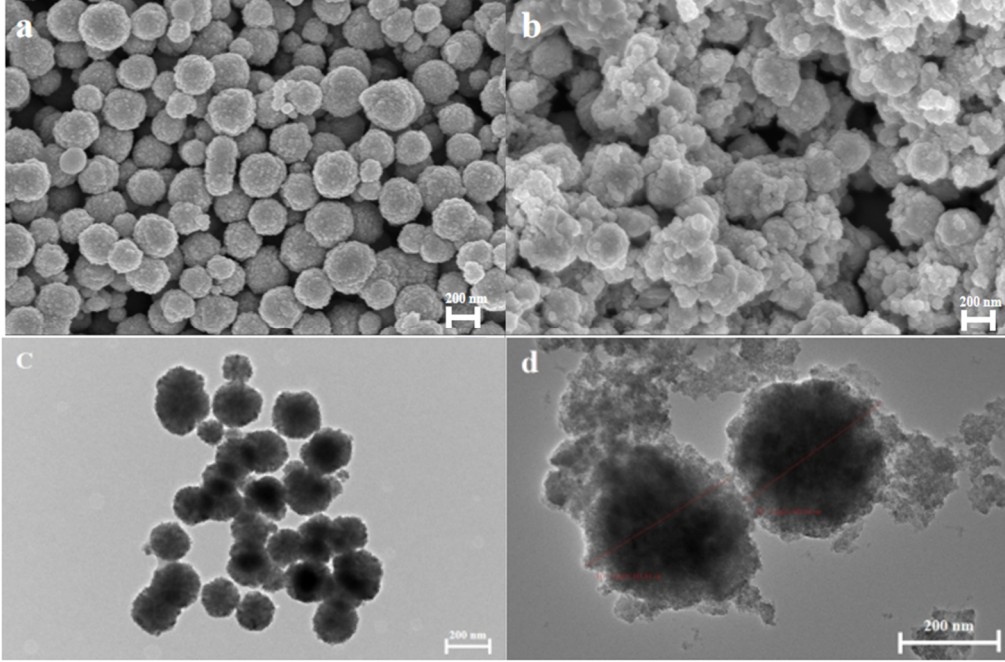

**Figure 1.** SEM and TEM images of $Fe_3O_4$ (**a,c**), $Fe_3O_4@ZnS@MPS$ (**b,d**).

### 3.1.2. XRD Analysis

Figure 2 shows the XRD patterns of $Fe_3O_4$, $Fe_3O_4@ZnS$, and MFNPs. In the XRD pattern, several diffraction peaks around $2\theta = 30.1°$, $35.6°$, $43.1°$, $53.7°$, $57.1°$, and $62.8°$ were observed for $Fe_3O_4@ZnS@MPS$, which could be assigned to the (220), (311), (400), (422), (511), and (440) planes of the cubic spinel crystal structure of $Fe_3O_4$, respectively. There were also three new diffraction peaks at around $2\theta = 28.9°$, $47.7°$, and $57.0°$, which could confirm the existence of ZnS crystals in the composites [24]. The position of the diffraction peak did not change when the quantum dots and sulfhydryl groups were modified, indicating that the $Fe_3O_4$ magnetic core did not undergo chemical or structural changes during the coating process. Figure 2 clearly shows that the spectral peaks are basically the same before and after functionalization, indicating that the crystal structure of $Fe_3O_4$ does not change during different functionalization processes [25,26].

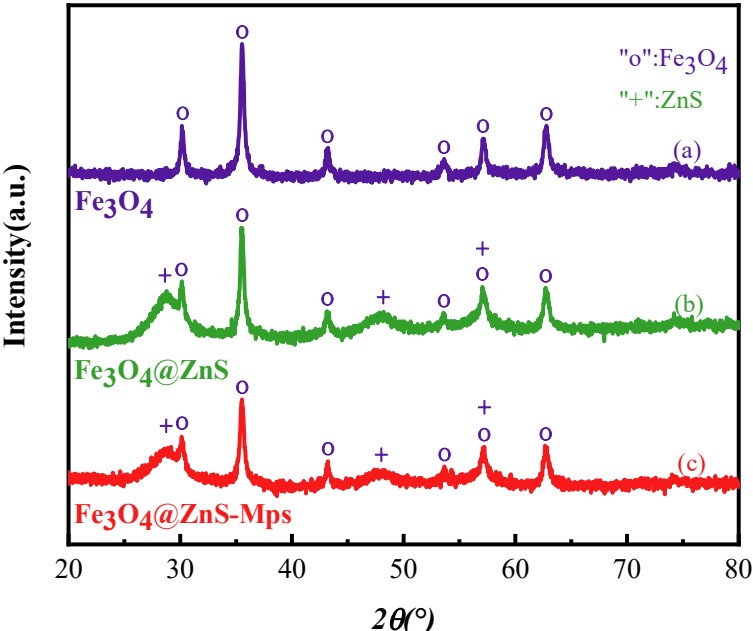

**Figure 2.** XRD patterns of (**a**) $Fe_3O_4$, (**b**) $Fe_3O_4@ZnS$, and (**c**) $Fe_3O_4@ZnS@MPS$ nanocomposites.

### 3.1.3. FT−IR Analysis

Figure 3 shows the infrared (IR) spectra of $Fe_3O_4$, $Fe_3O_4@ZnS$, $Fe_3O_4@ZnS@MPS$, and MPS. It can be seen from the infrared spectra of the magnetic microspheres that the strong absorption peak at 584 cm$^{-1}$ is related to the stretching vibration of the Fe–O bond. For $Fe_3O_4$ @ZnS magnetic fluorescent nanoparticles, the peaks corresponding to the stretching vibration of Fe–O and Zn–S appeared at 580 cm$^{-1}$ and 627 cm$^{-1}$. The stretching vibration of -SH in MPS corresponds to a peak at 2555 cm$^{-1}$. For MFNP nanoparticles, the Zn–S stretching vibration peak was obviously masked after sulfhydryl modification at 1018 cm$^{-1}$, and a new C–H peak appeared at 2922 cm$^{-1}$ [27]. The corresponding peak of the new S–O stretching vibration was 1042 cm$^{-1}$, while the peak at 1178 cm$^{-1}$ was assigned to the O=S=O symmetric stretching vibration. This can prove that the sulfhydryl group is bound to the surface of the microspheres and the ligand is successfully modified on the surface of the microspheres [28–31].

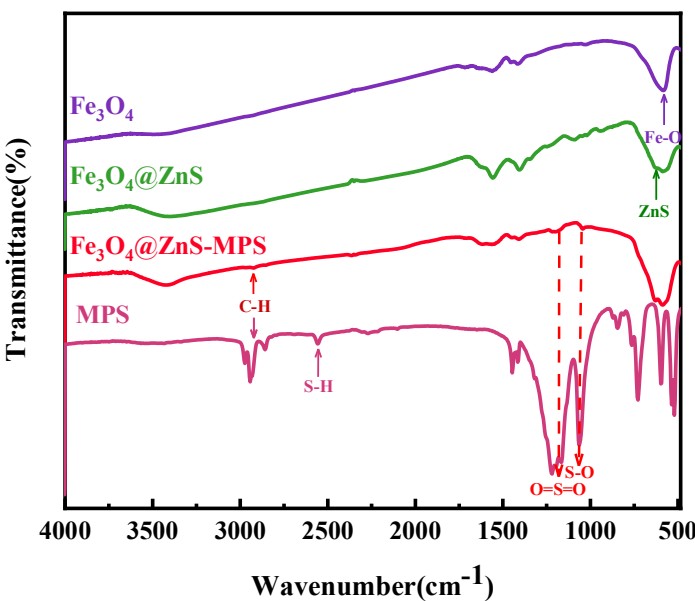

**Figure 3.** FT−IR spectra of commercial $Fe_3O_4$, $Fe_3O_4$@ZnS, $Fe_3O_4$@ZnS@MPS, and MPS.

### 3.1.4. XPS Analysis

The elemental composition of the $Fe_3O_4$@ZnS@MPS nanoparticle was explored by XPS analysis (Figure 4), The five peaks at 1019.2 eV, 709.8 eV, 529.6 eV, 285.6 eV, and 159.2 eV are composed of Zn 2p, Fe 2p, O 1s, C 1s, and S 2p, respectively, confirming the successful synthesis of $Fe_3O_4$@ZnS@MPS nanoparticles. Figure 4b shows that the binding energies of $Fe^{3+}$ $2p_{3/2}$ and $Fe^{3+}$ $2p_{1/2}$ are 710.8 and 725.0 eV, respectively, and the bimodal fitting of Fe 2p can obtain the binding energies of $Fe^{2+}$ $2p_{3/2}$ and $Fe^{2+}$ $2p_{1/2}$ at 708.7 and 721.6 eV, respectively. The results indicate that $Fe_3O_4$ exists in the nanoparticle. According to Figure 4c, the difference in binding energy between Zn $2p_{3/2}$ and Zn $2p_{1/2}$ is 22.5 eV, indicating that metallic Zn supported by the MFNPs mainly exists in the $Zn^{2+}$ valence state. Peaks at 159.5 eV and 160.7 eV (Figure 4d) are attributed to metal sulfides $S^{2−}$ ($2p_{3/2}$ and S $2p_{1/2}$) from ZnS, respectively [32,33].

### 3.1.5. Hysteresis Curve

The magnetic properties of the $Fe_3O_4$ and $Fe_3O_4$@ZnS@MPS MFNPs were studied using a VSM in an external magnetic field from −20,000 to +20,000 Oe, and results are shown in Figure 5. The saturation magnetization of $Fe_3O_4$ MNPs was 64.52 emu/g, and that of $Fe_3O_4$@ZnS@MPS nanocomposites was 47.09 emu/g. The magnetization in the nanocomposites is reduced due to the diamagnetic effect of the thick sulfhydryl-modified ZnS layer around the $Fe_3O_4$ MNPs. Ion redistribution of $Zn^{2+}$ may also have contributed to the decrease in the saturation magnetization of the composite. However, it still possesses typical superparamagnetism, which can meet the experimental requirements, and still shows excellent magnetic properties after removal of $Ag^+$ (inset figure) [11,34].

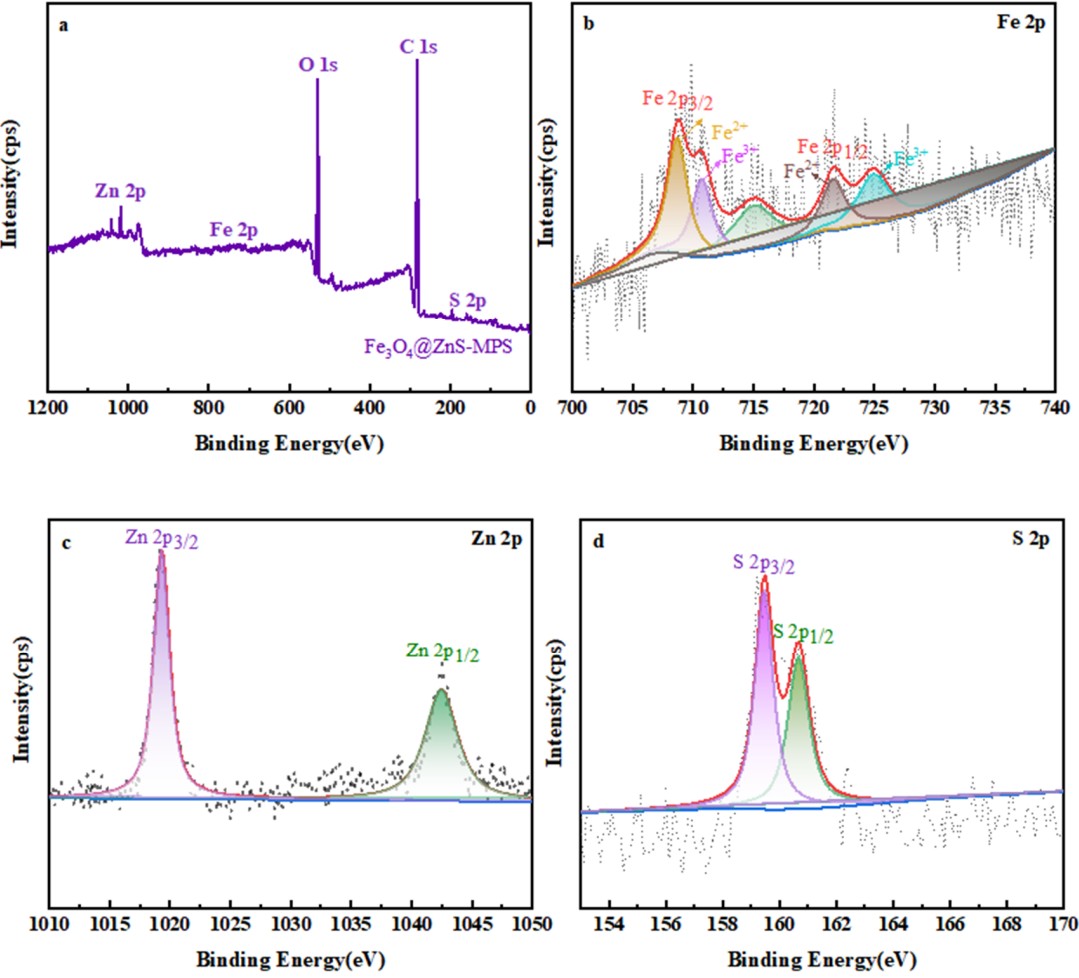

**Figure 4.** XPS analysis of Fe$_3$O$_4$@ZnS@MPS (**a**), high-resolution XPS spectra of Fe 2p (**b**), Zn 2p (**c**), S 2p (**d**) of Fe$_3$O$_4$@ZnS@MPS.

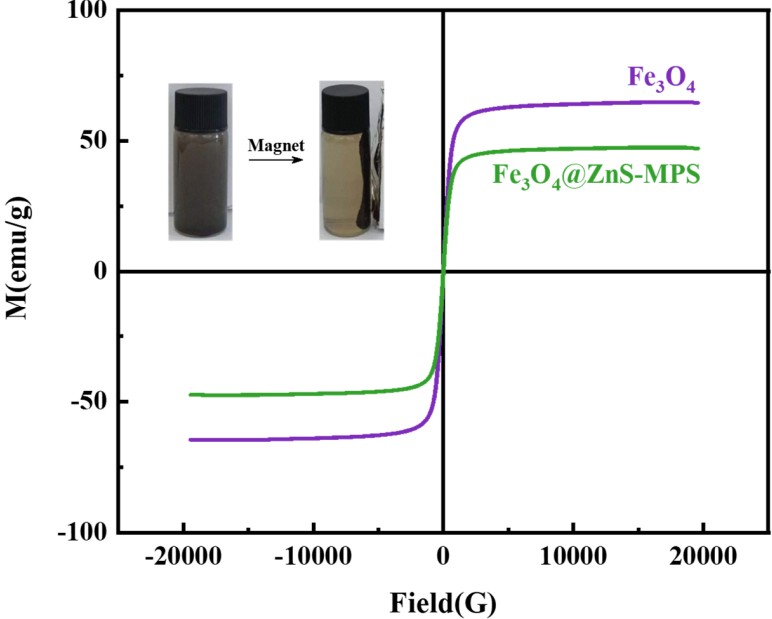

**Figure 5.** VSM measurements of Fe$_3$O$_4$ and Fe$_3$O$_4$@ZnS@MPS. Inset: The response of Fe$_3$O$_4$@ZnS@MPS to external magnetic field.

3.1.6. Thermogravimetric Analysis

In order to study the thermal properties of the samples, a thermal weight loss analyzer was used to analyze the samples. The thermal properties constitute an important index used to evaluate the hybrid materials, and differ with different synthesis methods. The TGA curves of MFNFs were measured at a heating rate of 10 °C/min under a nitrogen atmosphere, and the results are shown in Figure 6. As shown in Figure 6, the overall weight loss rate of the products is not very large, and the weight loss rate of $Fe_3O_4$ is only about 8%. After the modification of ZnS, its weight loss is reduced to 15%, and the extra weight loss is the water contained in ZnS. The weight loss rate reaches 18% after the modification of MPS, indicating that the polymer in the sample was removed. Therefore, the TGA analysis shows that the sulfhydryl group was modified on the surface of $Fe_3O_4$ magnetic microspheres.

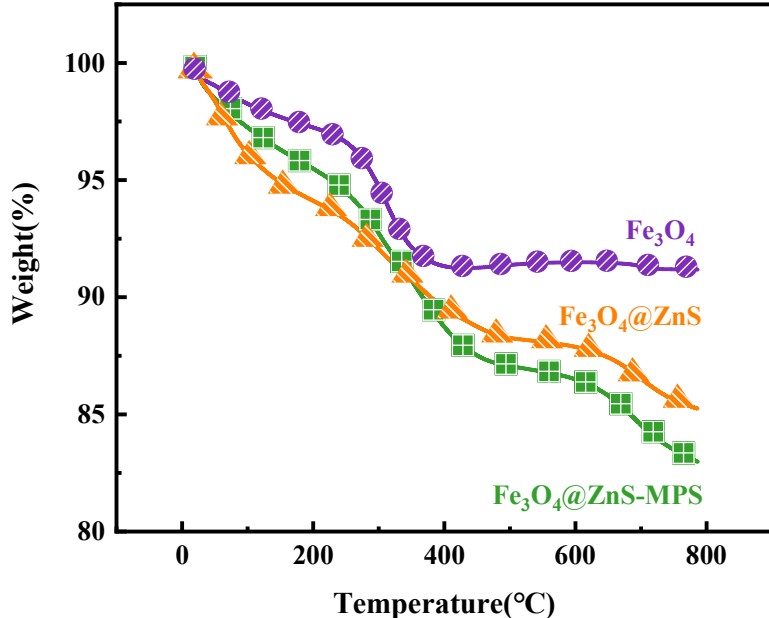

**Figure 6.** TGA curves of $Fe_3O_4$, $Fe_3O_4$@ZnS, and $Fe_3O_4$@ZnS@MPS.

Combined with the above data, it can be proven that the $Fe_3O_4$@ZnS@MPS magnetic fluorescent nanosensors (MFNPs) have been successfully synthesized.

*3.2. Detection Performance Study of MFNPs*

3.2.1. Performance Analysis of Magnetic Fluorescence Nanosensor MFNPs

Figure 7 shows the comparison of fluorescence characteristic spectra before and after the addition of $Ag^+$ to MFNPs. Figure 7 shows that the addition of $Ag^+$ can significantly quench the fluorescence intensity of MFNPs, and the inset shows the TEM images of MFNPs before and after the addition of $Ag^+$. It can be clearly seen that $Ag^+$ has been loaded and dispersed on the surface of the MFNPs. MFNPs have good dispersibility and are almost spherical. The complexation of $Ag^+$ with sodium 3-sulfhydryl-1-propane sulfonate on the surface of MFNPs leads to $Ag^+$ aggregation and obvious fluorescence quenching.

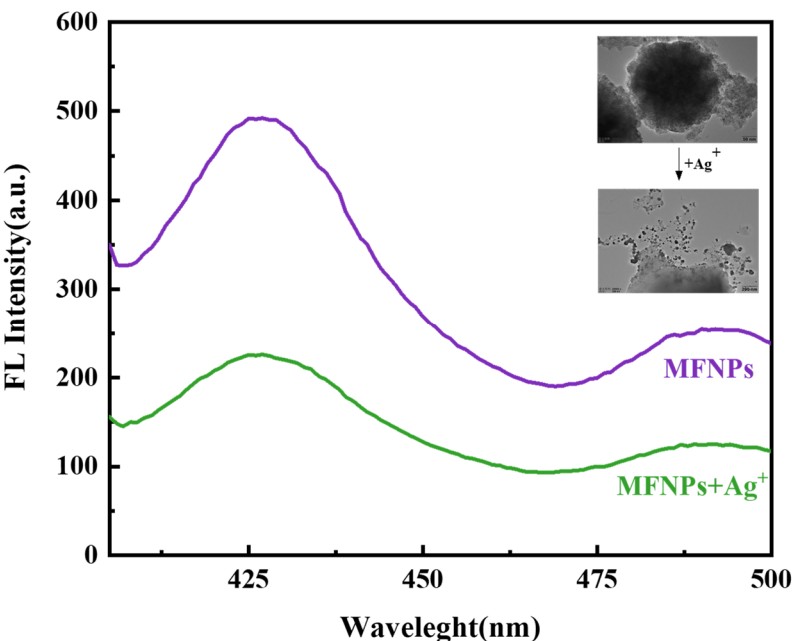

**Figure 7.** Comparison of fluorescence spectra of MFNPs before and after Ag$^+$ addition. Inset: TEM images of MFNPs before and after Ag$^+$ addition.

### 3.2.2. The Effect of pH on the Fluorescence of MFNPs

One of the crucial factors determining the sensor's capacity for detection is the pH level of the solution. As can be seen from Figure 8, the fluorescence intensity of the probe itself does not change greatly within the range of 4.8–9.0, and the degree of fluorescence intensity quenched by the addition of silver ions was not affected by the pH value. Therefore, we chose the common neutral liquid pH value of 7.0 as the experimental standard.

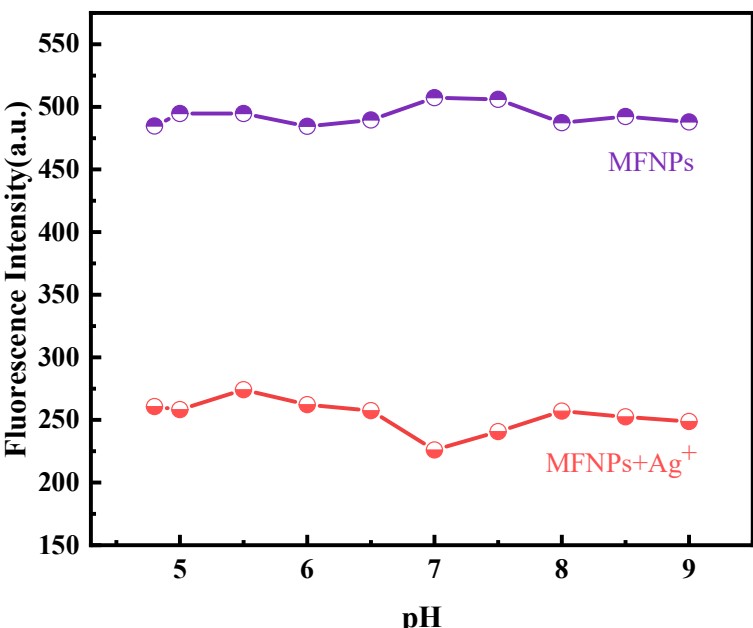

**Figure 8.** Fluorescence intensity of MFNPs at 425 nm in the absence and presence of Ag$^+$ at different pH.

### 3.2.3. The Response Range and Relation Curve of MFNPs to Ag$^+$

With the increase in the concentration of Ag$^+$, the fluorescence intensity gradually weakens in the range of 0–100 μM (Figure 9). There was a linear relationship between the concentration of Ag$^+$ and the fluorescence intensity I/I$_0$ = −0.00426x + 1.00934 (R$^2$ = 0.9967), where x is the concentrations of Ag$^+$ (μM), with the LOD of 7.04 μM. Therefore, the sensor of the presented invention can more accurately determine the content of Ag$^+$. Based on the data of the linear range and the detection limit, it can be seen that the probe showed good performance (Table 1).

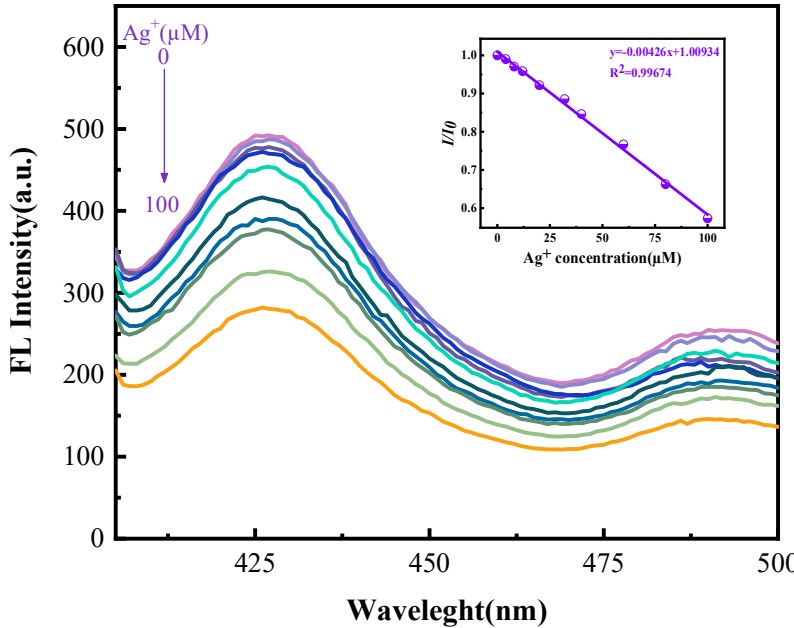

**Figure 9.** Emission spectra of MFNPs in the presence of increasing amounts of Ag$^+$ at room temperature. Inset: The curve of fluorescence intensity at 425 nm; the concentrations of Ag$^+$ are 0, 4, 8, 12, 20, 32, 40, 60, 80, 100 μM, respectively.

**Table 1.** Comparison of the present nanoprobe with other reported sensors for Ag$^+$ detection.

| Type of Sensor | Detection Range | LOD | Reference |
|---|---|---|---|
| Eu-MIL-61(Ga(OH)(btec)·0.5H$_2$O) | 0–1000 μM | 0.23 μM | Dalton Transactions, 46 (2017)875 [35] |
| Triphenylamine–thiophene-pyridinium | 5.0–9.0 μM | 3.6 μM | Talanta, 255(2023)124222 [36] |
| Nitrogen and bromine co-doped carbon dots | 0–6.0 μM | 0.14 μM | Spectrochimica Acta Part A: Molecular and Biomolecular Spectroscopy, 296(2023)122642 [37] |
| NPCl-doped carbon quantum dots | 15.89–27.66 μM | 26.38 μM | Analytica Chimica Acta, 1144 (2021) 1–13 [38] |
| Cholesteric chiral artificial receptor L5 | 2–20 μM | 0.13 μM | Microchemical Journal, 190(2023)108633 [39] |
| Fe$_3$O$_4$@ZnS@MPS | 0–100 μM | 7.04 μM | This work |

### 3.2.4. Particle Selectivity and Interference Ion Determination

High selectivity is a necessary condition for the sensor. Therefore, under the same conditions, the Ag$^+$ selectivity of the prepared magnetic fluorescent sensor was investigated by detection of the reaction of the relevant analytes. The results show that the fluorescence quenching effect of Ag$^+$ is the best. Although the other ions will experience weak quenching, this is obviously negligible compared with that of silver ions (Figure 10a). To further investigate the ability of the magnetic fluorescent sensor to recognize silver ions in the

presence of other metal ions, the anti-interference ability of the sensor was also investigated. When an equal amount of silver ions was added to an equal amount of other metal ions (400 μM $Ag^+$, $Co^{2+}$, $Ni^{2+}$, $Al^{3+}$, $Cu^{2+}$, $Zn^{2+}$, $Cd^{2+}$, $Fe^{3+}$, $Fe^{2+}$, $K^+$, $Ca^{2+}$, $Na^+$, $Pb^{2+}$, $Hg^{2+}$), the other ions did not affect the detection of silver ions by the magnetic fluorescent sensor. The bar graph clearly proves this point (Figure 10b); it is obvious that the common metal ions in the environment do not interfere with the qualitative and quantitative detection of silver ions by the particles.

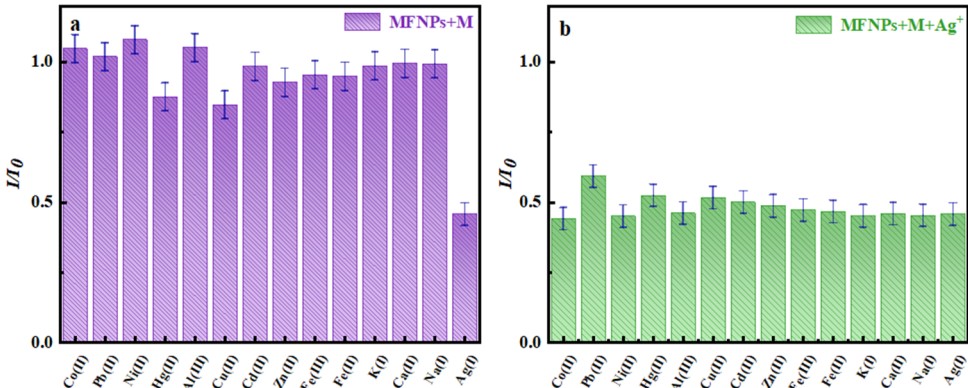

**Figure 10.** (**a**) Bar graph represents the ratio of fluorescence quenching of MFNPs in the presence of different metal ions (1, $Co^{2+}$; 2, $Pb^{2+}$; 3, $Ni^{2+}$; 4, $Hg^{2+}$; 5, $Al^{3+}$; 6, $Cu^{2+}$; 7, $Zn^{2+}$; 8, $Cd^{2+}$; 9, $Fe^{3+}$; 10, $Fe^{2+}$; 11, $K^+$; 12, $Ca^{2+}$; 13, $Na^+$; 14, $Ag^+$); (**b**) bar graph represents the ratio of fluorescence quenching of MFNPs upon the addition of $Ag^+$ (400 μM) to the solution containing other metal ions (400 μM, 1, $Co^{2+}$; 2, $Pb^{2+}$; 3, $Ni^{2+}$; 4, $Hg^{2+}$; 5, $Al^{3+}$; 6, $Cu^{2+}$; 7, $Zn^{2+}$; 8, $Cd^{2+}$; 9, $Fe^{3+}$; 10, $Fe^{2+}$; 11, $K^+$; 12, $Ca^{2+}$; 13, $Na^+$;14, Blank).

### 3.2.5. Removal of $Ag^+$

Adsorption capacity is considered to be one of the most important properties of nanomaterials, so adsorption tests were performed in conical flasks (100 mL) to determine the adsorption amount of $Ag^+$. The effect of the initial $Ag^+$ concentration on the adsorption efficiency is shown in Figure 11. The adsorption analysis showed that with the increase in the initial $Ag^+$ concentration, the adsorption capacity of MFNPs also gradually increased. This was due to the increase in the initial concentration, which improved the adsorption driving force and ion mass transfer rate, and the binding site on the adsorbents' surface was gradually occupied by $Ag^+$; thus, the adsorption capacity increased. The removal rate of $Ag^+$ in the solution gradually decreased after 100 μM, reaching 99.62%, which is because the amount of adsorbent in the solution is finite, and the adsorption site provided for $Ag^+$ is also finite. When the concentration of heavy metals is low, almost all $Ag^+$ can be combined with the adsorption site to achieve a higher removal rate, and then the adsorption gradually reaches equilibrium. Therefore, the adsorption effect of the adsorbent on the free metal ions in the solution is reduced, resulting in a lower removal rate. When the $Ag^+$ concentration changed from 300 μM to 600 μM, the removal rate decreased from 99.54% to 93.54%, and the adsorption capacity increased from 322.496 mg/g to 606.112 mg/g. In general, the optimal $Ag^+$ concentration at the intersection point of 400 μM could be obtained. Compared with other materials, the adsorption and transfer rates of this material are relatively high (Table 2).

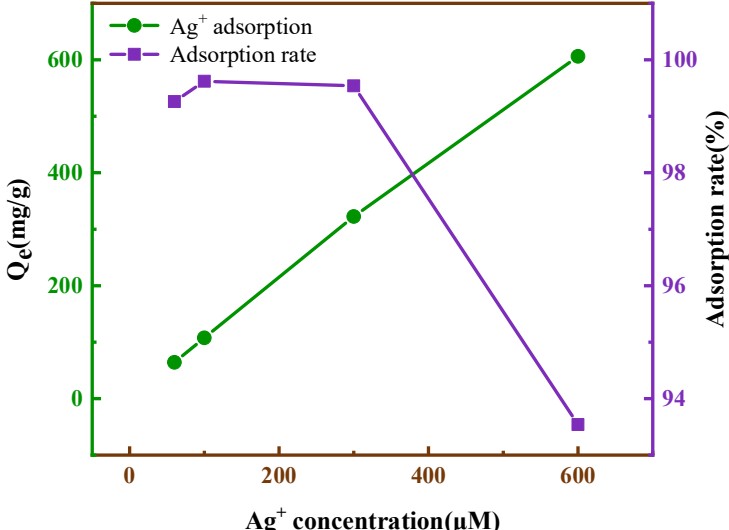

**Figure 11.** Effect of the initial concentration of Ag$^+$ on the adsorption capacity and removal efficiency of magnetic fluorescent nanoparticles.

**Table 2.** The adsorption capacity of reported adsorbents for Ag$^+$.

| Type of Sensor | Adsorption Capacity (mg/g) | Reference |
|---|---|---|
| Bifunctional polysilsesquioxane microspheres | 416.88 | Journal of Hazardous Materials 442(2023) 130,121 [40] |
| ZMC-MAH-TEPA (grafted magnetic zeolite/chitosan) | 70.12 | International Journal of Biological Macromolecules 222 (2022) 2615–2627 [41] |
| DMTD-AP(2,5-dimercap-to-1,3,4-thiadiazole modified apple pomace) | 196.9 | Sustainable Chemistry and Pharmacy 26 (2022) 100,621 [42] |
| L-PRL(o -phenanthroline-based polymer) | 325.8 | Chinese Chemical Letters 34 (2023) 107,485 [43] |
| MFNPs | 395.79 | This work |

*3.3. Infrared Spectroscopy Analysis before and after Complexation of MFNPs with Ag$^+$*

Infrared spectra of MFNPs before and after the addition of Ag$^+$ were analyzed using FT-IR, and the FT-IR spectra of MFNPs and MFNPs + Ag$^+$ complexes are shown in Figure 12. From the infrared spectra of MFNPs, the corresponding peaks of S = O telescopic vibration at 1042 cm$^{-1}$ were observed.

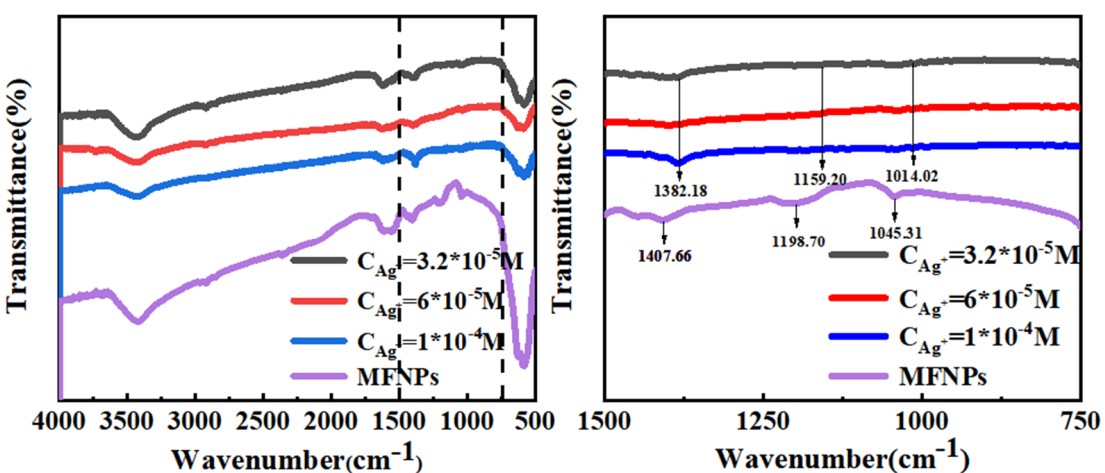

**Figure 12.** Infrared absorption spectra of MFNPs and inclusion complex MFNPs + Ag$^+$.

The peak shape of the absorption peaks at 1045.31 $cm^{-1}$, 1198.70 $cm^{-1}$, and 1407.66 $cm^{-1}$ changed, indicating that S–O, O=S=O, and C–O might play a key role in the synergistic effect with $Ag^+$. The result can be clearly observed in Figure 8; when the $Fe_3O_4$@ZnS-MPS interacted with Ag+, the tensile peak of methoxy (S–O) was changed from 1045.31 $cm^{-1}$ to 1014.02 $cm^{-1}$, while the characteristic absorption peak of O=S=O moved from 1198.70 $cm^{-1}$ to 1159.20 $cm^{-1}$ and the characteristic absorption peak of C–O moved from 1407.66 $cm^{-1}$ to 1382.18 $cm^{-1}$. The reason for the change in the infrared absorption peak of the sulfonic acid group might be that the oxygen atoms in the sulfonic acid group cooperated with $Ag^+$ to form a synergistic complexation. The oxygen atoms of the sulfonic acid group had a negative charge, indicating that the S–O and O=S=O group could adsorb $Ag^+$ by electrostatic interaction as $Ag^+$ laced electrons. Thus, it was speculated that the oxygen atom of the S–O and O=S=O group donated electrons to the $Ag^+$, which led to the transfer of charge and resulted in changes in the infrared spectrum [43].

### 3.4. The Fluorescent Detection of $Ag^+$ in Actual Samples

The practicability of the MFNPs for the detection of $Ag^+$ was demonstrated by analyzing real water samples. The real samples were selected from tap water, water from Songhua River in Jilin, and electrolysis waste water. First, suspended particles were removed from these water samples using centrifugal separation. Then, the supernatant was analyzed according to the previous procedure of $Ag^+$. Certain volumes of the supernatant and $Ag^+$ solutions were added to 200 µL of the $Fe_3O_4$@ZnS@MPS suspension, and the final volume was tuned to 2 mL by PBS buffer solution. As shown in Table 3, $Ag^+$ was not detected in the real water samples. The recovery of $Ag^+$ detection was in the range of 85.20%–105.30%, with the RSD (relative standard deviation) in the range of 3.18%–5.32%. The result indicated that the MFNPs had great potential for the detection of Ag+ in real water samples.

**Table 3.** The performance of the MFNPs in real water samples.

| Sample | Added (µM) | Measured (µM) | Recovery (%) | RSD (%) |
|---|---|---|---|---|
| | 1 | 0.926 | 92.60 | 3.18 |
| Tap water | 5.0 | 4.859 | 97.18 | 4.25 |
| | 10.0 | 10.321 | 103.21 | 3.86 |
| | 1 | 1.053 | 105.30 | 4.34 |
| River water | 5.0 | 4.987 | 99.74 | 3.89 |
| | 10.0 | 9.842 | 98.42 | 4.45 |
| | 1 | 0.852 | 85.20 | 5.32 |
| Electrolysis waste water | 5.0 | 4.320 | 86.40 | 4.96 |
| | 10.0 | 8.620 | 86.20 | 4.85 |

### 4. Conclusions

In summary, we present the design, synthesis, and performance of a novel magnetic fluorescent nanoprobe ($Fe_3O_4$@ZnS@MPS(MFNPs)) modified with 3-sulfhydryl-1-propane sodium for simultaneous detection and removal of $Ag^+$ from water solutions. Our proposed $Fe_3O_4$@ZnS@MPS sensor exhibits significant quenching fluorescence intensity ability and high selectivity for $Ag^+$. It was shown that the $Fe_3O_4$@ZnS@MPS fluorescent sensor was a good adsorbent for the removal of $Ag^+$, due to the aggregation and magnetic separation of the sensor. In addition, it could be applied over a wide pH range. The above results confirmed that this method is an inexpensive, simple, convenient, and extremely sensitive method that enables highly specific recognition for the simultaneous detection and removal of $Ag^+$ from aqueous solutions. This study provided a new idea for the determination and removal of $Ag^+$ and other heavy metals.

**Author Contributions:** Conceptualization, Y.G. and X.C.; methodology, X.Z.; software, X.C.; validation, Z.L., S.Y. and J.C.; formal analysis, P.X.; investigation, X.Z.; resources, Z.L.; data curation, X.C.; writing—original draft preparation, X.C.; writing—review and editing, X.Z.; visualization, X.Z.;

supervision, X.Z.; project administration, X.Z.; funding acquisition, Z.L. All authors have read and agreed to the published version of the manuscript.

**Funding:** This research was funded by X.Z. grant number (20220203020SF) And The APC was funded by Jilin Provincial Department of Science and Technology. This research was funded by J.C. grant number (51902125) And The APC was funded by National Natural Science Foundation of China. This research was funded by S.Y. grant number (22106051) And The APC was funded by National Natural Science Foundation of China.

**Data Availability Statement:** Z.X. and G.Y. designed the work. C.X. performed the experiment and data analysis.

**Acknowledgments:** We gratefully acknowledge financial support from Jilin Provincial Department of Science and Technology (20220203020SF), the National Natural Science Foundation of China (51902125, 22106051).

**Conflicts of Interest:** The authors declare no conflict of interest.

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
