# Peer review of "A Novel Magnetic Fluorescent Fe3O4@ZnS@MPS Nanosensor for Highly Sensitive Determination and Removal of Ag+"

_coatings, doi:10.3390/coatings13091557_

Round 1

Reviewer 1 Report

In “A novel magnetic fluorescent nanosensor Fe3O4@ZnS@MPS for highly sensitive determination and removal of Ag+” (ID coatings-2538663), authors report on the core-shell nanoparticles capable of ratiometric selective sensing of Ag+ and its removal. Authors present rigorous structural investigations as well as many interesting and practically important results, but related discussions have a significant room for improvement.

Major comments:

1) In the paper, authors analyze the sample deposited in one set of parameters only, and no motivation behind the choice of the synthesis parameters (reactants concentration, reactions times) was presented. Why were these parameters chosen? Were any other sets of parameters tested?

2) In the introduction, please provide the motivation behind the Ag+ removal discussed in Subsection 3.2.5.

3) Lines 128-129 indicate that "NPs have a (...) uniform distribution", however, these distributions are not presented. I suggest authors to carry out the analysis of the obtained images, to present a distribution of the particles and discuss it in more detail (for example, if the size of the NPs optimal for practical applications). Such analysis can also allow authors to make the conclusions on the size of the magnetite, ZnS and MPS components, which would be beneficial for the reproducibility of the study.

4) As O-Na bonds weren't observed by FTIR, I suggest authors to present and discuss the elemental composition assessed via XPS to prove that sodium is present in the structure.

5) Section 3.1.6: the variation in the desorption characteristics of Fe3O4 and Fe3O4@ZnS is attributed to more significant water presence in the latter. Shouldn't water desorption take place in 0-100 degrees range only, and, if the variation is explained by H2O only, shouldn't the desorption trend of Fe3O4 and Fe3O4@ZnS be relatively similar (only shifted after water evaporation) for the TGA range of 100-800 degrees?

6) Section 3.1.2: what is the origin of the observed luminescence peaks? Please supplement the discussion.

7) Section 3.1.2: Surprisingly, mechanism responsible for the sensing capabilities of the presented material is not analyzed in present study. Why does quenching take place after the Ag+ interaction with MPS? How is it related to the structure disruption (inset of Fig. 7)? Please discuss this effect in more detail.

8) The inset of Fig. 7 is misleading. First, the scale of the TEM images differs by 4 times. As I see it, bottom subfigure indicates large particle/agglomerate, but the formation of such structures is not discussed in the manuscript. Second, I don’t think that TEM allows to see the dynamics of the process of Ag+ and MFNP interaction, as this process doesn’t take place on TEM grid, and samples transfer to the grid changes the position of the fragments. Please revise the figures and discuss the revised inset in more detailed way.

9) If the sensing capabilities of the structure are explained by only the Ag+ interaction with outer layer of the structure (MPS), what is the benefit of Fe3O4@ZnS-MPS material?

10) As reversibility of the materials performance is crucial for its practical application, consider adding the discussion on if sensing and sorption properties of the material are reversible.

11) In the top subfigure of the inset of Fig. 7, 300-400-nm-sized particle is presented. However, in Section 3.1.1 (fig. 1b,d), 200-nm-sized particles are reported, and their "regular and uniform distribution" is claimed. How is that possible?

12) Section 3.2.4: what is the suggested mechanism of the sensor’s selectivity?

Minor comments:

13) In the Section 2, please provide the experimental details regarding Ag removal.

14) In the Introduction and/or main text, define "MPS".

15) Line 39, what kind of "cell" do you mean?

16) In the Introduction, it was communicated that organic and oxide stabilizers are used to encapsulate magnetite NPs. However, the purpose of the Fe3O4 encapsulation into ZnS was not disclosed. Consider supplementing the manuscript.

17) Lines 97-98, "After the item was attracted by a magnetic force": please provide more detail, what was the purpose of this step, was electromagnet or permanent magnet used? What was the overall setup?

18) Section 2.4: the parameters of analysis should be discussed in more detail: what was the energy if incident TEM and SEM electrons, what excitation wavelengths were used for XRD and XPS, how was XPS calibrated? How were the samples prepared for TEM?

19) Lines 121-122, what solutions were used as sources of ions, how were they dissolved?

20) In fig. 1d, the red markings are not visible and not discussed in the text.

21) Lines 139-143, appropriate reference regarding the magnetite and ZnS diffraction pattern should be provided.

22) XRD peaks related to ZnS are significantly wider than the ones related to Fe3O4. Does it mean that ZnS component is amorphous or its crystallites are smaller than the ones of magnetite?

23) In the X axis title of Fig. 2, "/" is redundant.

24) For FTIR spectra of Fe3O4, the intensive line at ~700 cm-1 is not attributed to any kind of bonding, as well as the less intensive bands in 700-900 cm-1 range. I assume that the line at 700 cm-1 is attributed to the maghemite [635 cm-1 in https://link.springer.com/article/10.1007/s10973-016-5393-y], and its emergence is related to the magnetite degradation in ambient air; therefore, I suggest discussing this aspect in more detail.

25) Is a wide band at ~3400 cm-1 attributed to the vibrations of the O-H bonds of adsorbed water [https://doi.org/10.3390/jcs7040156]? I Does its intensity correlate with the desorption intensity (data reported in in Subsection 3.1.6)? Please supplement the discussion.

26) Additionally, I suggest authors to supplement the FTIR-related discussion regarding the position of C-H bonds. It was reported that the C-H positions of chained (more generally, aliphatic) carbon structures differ from the ones of aromatic structures [https://doi.org/10.3390/jcs7040156 and refs. within]. Therefore, the C-H positions can provide the insight on if carbon substructure is indeed comprised of the linear polymeric fragments or if they are crosslinked.

27) Line 168: 282.7 eV C1s position is unconventional for sp2/sp3 hybridized carbon structures, while sp-bonding is unlikely to be observed for the investigated structures. Were the survey spectra really measured with 0.1 eV precision? Please revise the data. For me, 284-286 eV C1s position would be more believable.

28) Please provide more details regarding the high-resolution XPS processing: how were the spectra fitted, what kind of baseline was subtracted? For me, it is unclear what do the grey and blue "shadows" and the straight lines in the Fig. 4(b, c, d) mean: consider providing a legend.

29) In the caption of Fig. 5, please communicate what is shown in the inset.

30) In Fig. 8, luminescence of what line is presented? Do authors analyze a peak amplitude or area below the peak?

31) Line 233, please communicate what does “x” mean (Ag concentration?), what are its measurement units.

32) In Table 1, references are out of order (Ref. 37 is placed after Ref. 32).

33) The letters and numbers in Fig. 4 and Fig. 12 seem distorted. In Fig. 10, tick labels of X axis are distorted, as well as the Y axis title. Please revise.

34) Tables 1 and 2 are not referenced in the text.

35) In fig. 12, percentage of transmittance is not shown (tick labels of the Y axis are missing). The measurement units of the Ag concentrations in the legend are should be added as well.

36) In Section 3.4, what do you mean by “recovery”?

37) Prior to Table 3, define RSD.

38) Line 318, why is the studied material named “Fe3O4” in the conclusions?

39) Please define the abbreviations related to the literature data used in the Tables 1,2.

40) Line 11, "cooperated with Ag+ in metal ions" phrase is unclear.

41) Line 15, "MFNPs were greatly specific recognition" phrase should be revised

42) Line 38, what do you mean by "select higher 3D Spaces"?

43) Line 46, "iron oxide (Fe3O4) particularly versatile": verb is missing.

44) Line 80 "Poly tetra fluoroethylene" should be written together (in one word).

45) Lines 203-204, "sulfhydryl group has been modified to Fe3O4 surface"

46) Line 217, what do you mean by "sensor aggregation "?

47) Line 270-271, “which is because the amount of adsorbent in the solution is certain, and the adsorption site provided for Ag+ is also certain”, did you mean “finite/constant/fixed”?

48) Lines 307-308, “according to the previous procedure of Ag+” phrase should be revised.

Reviewer 2 Report

The manuscript ID coatings-2538663 mainly presents a study about synthesis and standard characterization of a particular magnetic fluorescent nanoprobe (Fe3O4@ZnS@3-sulfhydryl-1-propane sodium) used to determine the content of Ag+ in liquid samples. Please see below a list of comments for the authors:

1. Fluorescence is highly dependent on area and direction of irradiation, please describe how were determined these parameters to guarantee reproducibility and effective comparison between measurements.

2. Is there an anisotropic response of the magnetic nanoparticles in respect to the direction of the magnetic field?

3. If possible, please substitute fluorescence measurements by quantum yield measurement to see the efficiency of the emission of the sensor proposed.

4. The aim of the work should be better justified highlighting what this work adds to literature.

5. It is not clear how were selected the parameters for the preparation of the samples.

6. The authors are invited to describe some perspectives about the use of advanced techniques like machine learning or multiphotonic effects to improve the sensing performance of the system proposed. You can see for instance: https://doi.org/10.3390/bios12090710

7. Advantages and disadvantages of the proposed Fe3O4@ZnS@MPS in respect to other elements must be discussed within the report. You can see for instance: https://doi.org/10.1016/j.matchemphys.2021.124474

8. How is the influence of the size of the particles described in figure 1 over the main findings?

9. Error bar in experimental data must be incorporated.

10. The presentation of the citations in collective form could be split in order to better describe the importance of the information associated with the references selected to be part of the introduction of the topic.

A proofreading is suggested

Round 2

Reviewer 1 Report

Authors of “A novel magnetic fluorescent nanosensor Fe3O4@ZnS@MPS for highly sensitive determination and removal of Ag+” (ID: coatings-2538663) have considerably improved the manuscript throughout the revision. However, authors haven’t made all of the necessary adjustments, and their answers weren’t sufficiently detailed, therefore I recommend another round of revision of this manuscript.

Comments:

11)      Lines 72-73, “other heavy metal ions with multifunction of enrichment, detection and separation”: what kind of “enrichment” is meant? Please discuss in a more detailed way why it is a solution to some actual problem.

22)      Line 112, please indicate if “CAE” means “Constant Analyzer Energy”.

33)      In Fig. 4, please add the legend.

44)      Line 264, please indicate the measurement units of “x” in “I/I0 = −0.00426x + 1.00934” equation (µM?).

55)      In the reply to Q.4 authors claim that “O-Na bond is not used, and there is no O-Na bond.” However, this type of bonding is indicated in Scheme 1, and it apparently exists in the MPS (Sodium 3-mercaptopropanesulphonate, https://pubchem.ncbi.nlm.nih.gov/compound/Sodium-3-mercaptopropanesulphonate). How can authors explain this discrepancy? Why Na was not observed by XPS, was its concentration below the sensitivity limit?

Reviewer 2 Report

The authors have successfully clarified most of the points raised in the review stage. The results are interesting and in my opinion, the main findings worth publication in present form.

A proofreading is suggested

Author Response

Received.